# Genome assembly with *in vitro* proximity ligation data and whole-genome triplication in lettuce

Sebastian Reyes-Chin-Wo[1,*], Zhiwen Wang[2,*,†], Xinhua Yang[2,*], Alexander Kozik[1], Siwaret Arikit[3,†], Chi Song[2], Liangfeng Xia[2], Lutz Froenicke[1], Dean O. Lavelle[1], María-José Truco[1], Rui Xia[4], Shilin Zhu[2], Chunyan Xu[2], Huaqin Xu[1], Xun Xu[2], Kyle Cox[1], Ian Korf[1,5], Blake C. Meyers[3,4] & Richard W. Michelmore[1,5,6,7]

Lettuce (*Lactuca sativa*) is a major crop and a member of the large, highly successful Compositae family of flowering plants. Here we present a reference assembly for the species and family. This was generated using whole-genome shotgun Illumina reads plus *in vitro* proximity ligation data to create large superscaffolds; it was validated genetically and superscaffolds were oriented in genetic bins ordered along nine chromosomal pseudomolecules. We identify several genomic features that may have contributed to the success of the family, including genes encoding *Cycloidea*-like transcription factors, kinases, enzymes involved in rubber biosynthesis and disease resistance proteins that are expanded in the genome. We characterize 21 novel microRNAs, one of which may trigger phasiRNAs from numerous kinase transcripts. We provide evidence for a whole-genome triplication event specific but basal to the Compositae. We detect 26% of the genome in triplicated regions containing 30% of all genes that are enriched for regulatory sequences and depleted for genes involved in defence.

[1] UC Davis Genome Center, Davis, California 95616, USA. [2] BGI Shenzhen, Shenzhen 518083, China. [3] Delaware Biotechnology Institute, University of Delaware, Newark, Delaware 19711, USA. [4] Donald Danforth Plant Science Center, 975 North Warson Road, St Louis, Missouri 63132, USA. [5] Department of Molecular & Cellular Biology, UC Davis, California 95616, USA. [6] Department of Plant Sciences, UC Davis, California 95616, USA. [7] Department of Medical Microbiology & Immunology, UC Davis, California 95616, USA. * These authors contributed equally to this work. † Present addresses: PubBio-Tech, Wuhan 430070, China (Z.W.); Rice Science Center, and Department of Agronomy, Faculty of Agriculture, Kasetsart University, Kamphaeng Saen, Nakhon Pathom 73140, Thailand (S.A.). Correspondence and requests for materials should be addressed to R.W.M. (email: rwmichelmore@ucdavis.edu).

The Compositae (also known as Asteraceae) is the most successful family of flowering plants on earth in terms of number of species and diversity of habitats colonized[1]. The family is thought to have originated in the mid-Eocene (45–49 Myr) and expanded greatly during the Oligocene (28–36 Myr)[2,3]. It encompasses 1,620 recognized genera and at least 23,600 species, constituting approximately 10% of all angiosperms[1]. They are easily recognizable by their characteristic compound inflorescences that comprise many true flowers. This cosmopolitan family is present in diverse habitats; Compositae species are successful colonizers of disturbed habitats and thrive in a range of extreme environments including deserts, tundra and salt flats[1,4]. Although it does not contain any of the top six food crops, the family includes many important edible, medicinal, noxious and invasive species[5,6]. Over 200 species have been domesticated for a wide variety of uses. In aggregate, over 27 million hectares are planted worldwide to Compositae crops[7], of which the two most important are lettuce and sunflower. Medicinal species include *Artemesia* spp. and Echinacea. Many ornamental species with their showy flowers belong to the Compositae. In addition, numerous troublesome weeds such as star thistle are also Composites. Only two draft sequences of Compositae species have been published so far. One is of the small 335 Mb genome of horseweed (*Conyza canadensis*)[8]; the second is of the 1 Gb genome of globe artichoke (*Cynara cardunculus*)[9].

Lettuce (*Lactuca sativa* L.) is an important vegetable crop species and ranks as one of the top 10 most valuable crops in the USA with an annual value of over $2.4 billion[10]. *L. sativa* is diploid with $2n = 2x = 18$ chromosomes and an estimated genome size of 2.5 Gb (refs 11,12). Generation of a comprehensive reference genome was challenging due to its size and high repeat content. We sequenced and assembled the genome of *L. sativa* using a variety of approaches that included a wide range of mate-pair libraries and *in vitro* proximity ligation to generate large superscaffolds based on long-range contact frequencies detected between scaffolds[13]. *In vitro* proximity ligation is an application of the chromosome conformation capture technologies to aid genome assembly[14]; this approach has been reported for animal but not for plant genomes[15,16]. The assembly was validated genetically and superscaffolds were arranged in genetic bins ordered along the nine chromosomal linkage groups. The resulting assembly is one of the more complete for any plant species reported so far, particularly for a genome larger than 2 Gb with a high repeat content, and provides the first high-quality, comprehensive reference genome for analysis of the Compositae family.

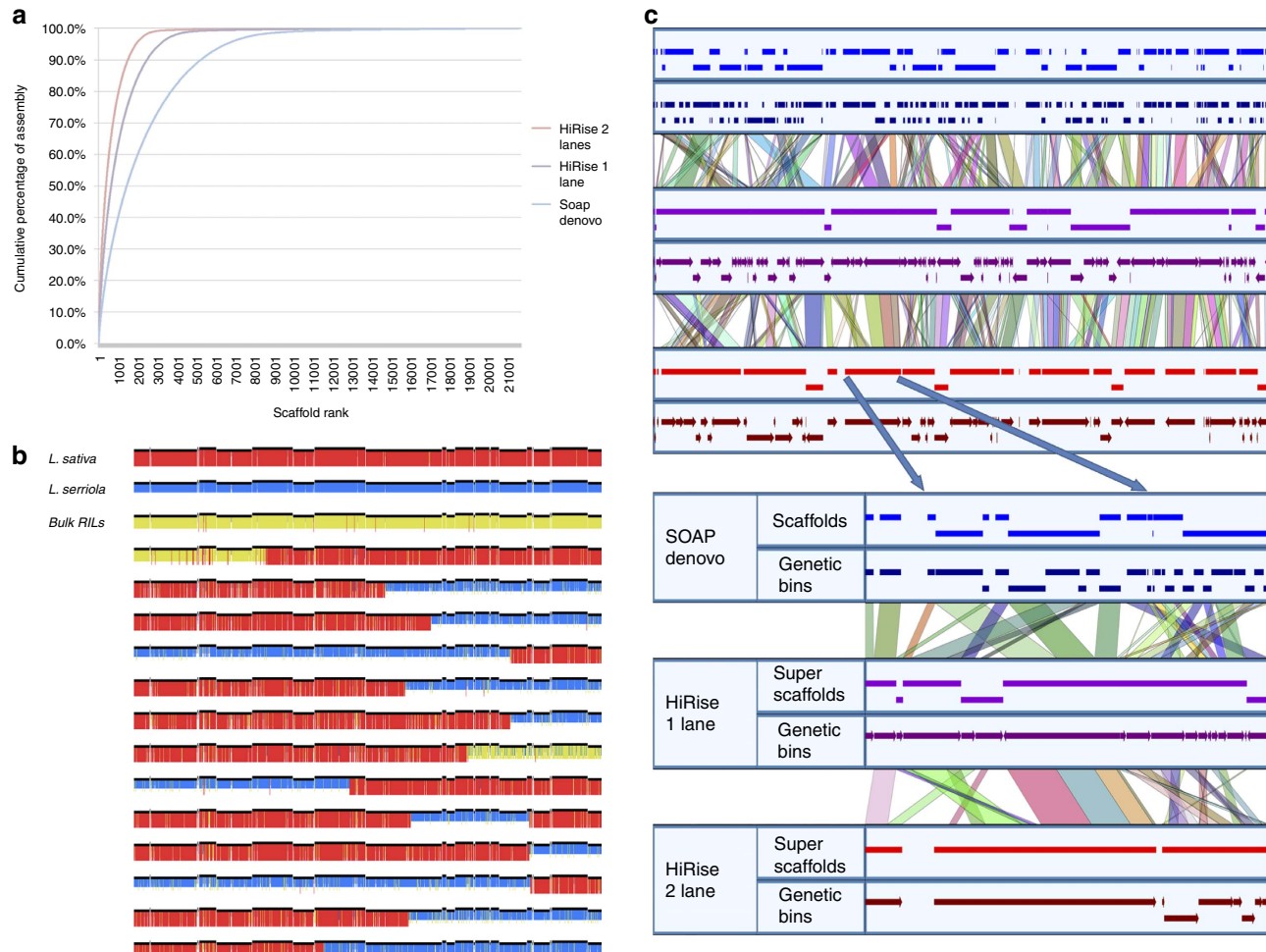

**Figure 1 | Improvement of the *L. sativa* cv. Salinas genome assembly after scaffolding with HiRise.** (**a**) Higher contiguity of the genome assembly by scaffold using HiRise as a function of an increase in the sequence size and proportion of large sequences. (**b**) Genotype calls for RILs with crossovers across the longest HiRise superscaffold. Red bars represent *L. sativa* alleles, blue bars represent *L. serriola* alleles and the yellow bars represent heterozygotes. Alternating discontinuities in the black line of top of the genotype represent joins between SOAPdenovo scaffolds. (**c**) Ordering, orientation and incorporation of additional scaffolds spanning the Major Resistance Cluster 2 (ref. 31) in the two HiRise assemblies. Expanded view shows a single super scaffold that organized four genetic bins into a single unit.

## Results

**Genome sequencing and assembly.** A whole-genome shotgun strategy was used to sequence and assemble the genome of *L. sativa* cultivar Salinas from Illumina short reads. A total of 198.5 Gb Illumina paired-end and mate-pair reads were generated from seven libraries of different fragment sizes (170 bp to 40 kb; Supplementary Table 1). After filtering, this provided 72.5-fold coverage of the 2.7 Gb genome as estimated by K-mer analysis, making this one of the larger plant genomes assembled so far[17]. An additional 47 Gb of paired-end reads from gene space libraries[18] were used when filling gaps. This initial SOAPdenovo assembly consisted of 153,952 contigs and 21,686 scaffolds greater than 1 kb with the largest scaffold being 3.1 Mb. The $N_{50}$s of contigs and scaffolds were 12 and 476 kb, respectively. The mean size of gaps in the scaffolds was 1.3 kb (Supplementary Table 2). The Chicago library data (*in vitro* proximity ligation) scaffolded with the HiRise software pipeline (Dovetail Genomics LLC, Santa Cruz, CA, USA) increased the contiguity of scaffolds considerably (Supplementary Table 3 and Fig. 1a). The final HiRise assembly decreased the 21,686 scaffolds to 11,474 superscaffolds and increased the $N_{50}$ from 476 to 1,769 kb; 50 and 90% of the genome is represented in only 385 and 1,520 superscaffolds, respectively (Table 1 and Supplementary Table 2). The largest superscaffold is 12.2 Mb and contains 27 SOAPdenovo scaffolds (Fig. 1b). The total length of the assembly is 2.38 Gb, covering ~88% of the estimated genome size of *L. sativa*. This assembly represents the gene space well; CEGMA[19] analysis found 97.6% matches to the set of 248 Conserved Eukaryotic Genes (CEGs[19]) (Supplementary Table 4) and all 357 Ultra-Conserved Orthologs

from *Arabidopsis thaliana* were found using TBLASTN (Supplementary Table 4). In addition, 96.6% of the 80,727 *L. sativa* expressed sequence tags (ESTs) in NCBI could be aligned to the genome assembly at >80% identity and >50% coverage (Supplementary Table 4).

**Validation and anchoring of the assembly to linkage groups.** To validate the assembly and generate chromosomal pseudo-molecules for *L. sativa*, we sequenced the gene space of 99 $F_7$ recombinant inbred lines (RILs) from an interspecific cross of *L. sativa* cv. Salinas × *L. serriola* acc. US96UC23 (ref. 20). Haplotypes could be assigned to 12,023 of the larger scaffolds. The SOAPdenovo assembly was validated using both genetic and HiRise information. Only 171 (0.78%) of the scaffolds were identified as chimeric based on the presence of indicative population-wide switches in genotype calls occurring at the point of misjoin (Supplementary Fig. 1). Another 74 (0.34%) were identified as chimeric based in discrepancies in the contact frequency along the scaffold. In total, only 245 (1.13%) scaffolds were chimeric and had to be split reflecting the high quality of the initial SOAPdenovo assembly.

A total of 9,140 scaffolds of the *L. sativa* assembly could be clustered into nine chromosomal linkage groups and then mapped into genetic bins ordered along each chromosomal linkage group (Supplementary Fig. 2 and Supplementary Data 2). The groups were named and oriented consistent with the previously reported, ultra-dense map[20] using genic sequences common to both maps. As expected, there was almost perfect colinearity between the map generated from segregation data resulting from hybridizations to the Affymetrix GeneChip and the map generated from the sequence data. Out of a total of 9,140 mapped scaffolds, less than 1% had inconsistent positions in the two maps; these inconsistencies may have been due to mapping of paralogs (Supplementary Fig. 3).

The combined genetic and HiRise[13] data provided further validation of the assembly, captured an additional 3,638 previously unmapped scaffolds that encompassed 49.5 Mb and refined the order and orientation of scaffolds in each genetic bin (Fig. 1c). Genetic data identified only 24 (0.21%) misassembled superscaffolds. After splitting chimeric scaffolds, nine chromosomal pseudomolecules were generated and displayed using GBrowse (Supplementary Fig. 4) (http://lgr.genomecenter.ucdavis.edu) with 3,138 superscaffolds encompassing 2.30 Gb (Supplementary Data 3) (96.6% of the assembly). The HiRise data reduced the number of sequences from an average of 6.26 scaffolds per bin to 2.25 superscaffolds per bin and resulted in reorganization and reorientation of scaffolds within genetic bins (Fig. 2, Track a and Supplementary Data 4). Of 3,112 superscaffolds that contained mapped fragments, 2,325 (74.7%) were located within a single genetic bin or had only a single mapped marker and so could not be oriented. The remaining 787 superscaffolds spanned on average 1.7 cM. Orientation could be determined for 859 superscaffolds based on crossovers within them (Fig. 1b); this accurately placed 1.5 Gb (63%) of the assembly in the precise location and orientation. Such improvements are exemplified by the Major Resistance Cluster 2 region where the number of sequences was decreased by a third for this complex region that contains large numbers of paralogs encoding candidate-resistance proteins (Fig. 1c). This improvement was also evident in the analysis of synteny on the genomic scale. We were able to detect twice the number of syntelogs in the final chromosomal pseudomolecules based on HiRise superscaffolds (9,325 syntenic hits) compared to the chromosomal pseudomolecules build with the SOAPDenovo scaffolds (4,038 syntenic hits) due to a significant improvement in

**Table 1 | Assembly statistics for the genome of *L. sativa* cv. Salinas.**

| Genome assembly metrics | SOAPDenovo | HiRise (2 lanes) |
|---|---|---|
| *Contigs* | | |
| N50 (size/number) | 36 kb/21,116 | — |
| Largest | 253 kb | — |
| Total size | 2.21 Gb | — |
| Total number | 153,952 | — |
| *Scaffolds* | | |
| N50 (size/number) | 476 kb/1,445 | 1.8 Mb |
| N90 (size/number) | 118 kb/5,237 | 360 kb/1,520 |
| Largest | 3.1 Mb | 12.2 Mb |
| Total size | 2.38 Gb | 2.38 Gb |
| Total number | 21,686 | 11,474 |
| **Genome annotation*** | | |
| | Family | Total Length |
| Transposable elements | Retroelements | 1.5 Gb (61.5%) |
| | DNA elements | 29.5 Mb (1.2%) |
| | MITEs | 103.7 Mb (4.4%) |
| | Others | 115.3 kb (<1%) |
| | Unknown | 152.9 Mb (6.3%) |
| | Total | 1.8 Gb (74.2%) |
| | Type | Copies |
| Non-coding RNA | rRNAs | 2,587 |
| | tRNAs | 1,347 |
| | Predicted miRNAs | 483 |
| | Detected miRNAs | 86 |
| | snRNAs | 1,514 |
| Protein coding genes | Total number | 38,919 |
| | Annotated transcripts | 31,348 |
| | Average CDS length | 1.05 kb |

MITE, Miniature Inverted-Repeat Transposable Elements.
*Annotation provided for HiRise assembly.

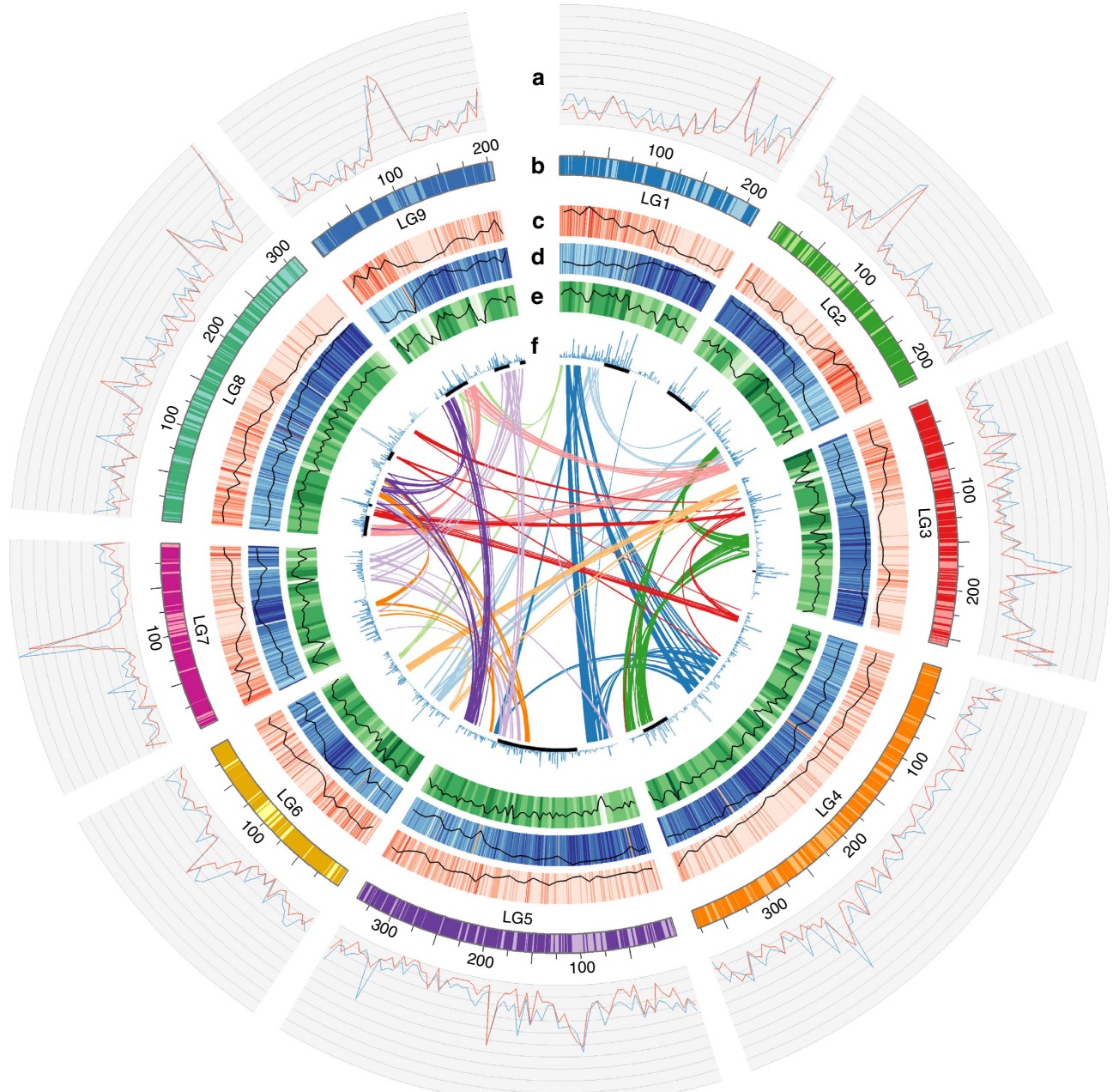

**Figure 2 | Overview of the *L. sativa* cv. Salinas genome.** (**a**) Number of scaffolds in 1 Mb intervals indicating the greater contiguity with the HiRise analysis. Blue, SOAPdenovo scaffolds; red, HiRise superscaffolds. (**b**) Chromosomal pseudomolecules. Dark areas indicate the 63% of the genome that is positioned and oriented accurately. (**c**) Gene density (in 1 Mb windows). (**d**) Repeat density (in 1 Mb windows). (**e**) Density of single-nucleotide polymorphisms used for genetic map construction (in 1 Mb windows). (**f**) Size of tandem gene arrays. Black blocks underneath show MRC regions[31]. The coloured lines in the centre show links between syntenic blocks of at least five genes derived from the most recent whole-genome triplication.

gene placement. Telomeric sequence arrays were detected within the terminal bins of seven pseudomolecules. Five chromosomes show single regions of elevated scaffold density on a genetic basis indicative of the likely position of the centromeres (Fig. 2, Track a).

**Transposon content and annotation.** Through a combination of approaches we found that 1.8 Gb of the 2.38 Gb (74.2%) assembled genome of *L. sativa* comprised repetitive elements (Table 1). The most abundant elements were long terminal repeat retrotransposons (LTR-RT), specifically the *Gypsy* (33.9%) and *Copia* (24.9%) subfamilies. Besides the main group of LTR-RT elements 4.4% of the genome was annotated as Miniature Inverted-Repeat

Transposable Elements and 1.2% as DNA elements, the remaining was classified into other repeat families or could not be assigned (Supplementary Fig. 4 and Supplementary Data 5). Repeat elements were distributed over all of the chromosomes; several chromosomes had internal regions with elevated densities of repeats, possibly indicative of centromeres but no major regions of heterochromatin were evident (Fig. 2, Track d).

**Gene annotation.** A high confidence gene set of 38,919 gene models with good protein or EST support was constructed for *L. sativa* by merging gene models from different prediction pipelines (see Methods). These gene models have average coding lengths of 1.05 kb and 4.5 exons per gene (Supplementary Data 6),

similar to those in other sequenced plant genomes (Supplementary Fig. 5). The average intergenic distance of 39.5 kb places *L. sativa* between *Nicotiana tomentosis* and *Capsicum annum* consistent with a direct correlation between intergenic distance and genome size (Supplementary Fig. 6). Out of the total number of predicted genes, 29,681 (76.27%) genes had similarity to *Arabidopsis* TAIR10 annotations (Supplementary Data 6) and 28,951 (74.3%) were annotated using InterProScan5[21] (Supplementary Data 7 and 8). Annotation using KEGG[22] database yielded information for 7,553 *L. sativa* gene predictions (Supplementary Data 9). The combined data sets provided functional annotation for 31,348 (80.5%) gene models.

**Small RNAs in lettuce and their target transcripts**. To identify microRNAs, loci generating phased, secondary siRNAs and other small RNAs, we sequenced 10 libraries each of small RNAs and uncapped or cleaved mRNAs ('PARE' libraries[23]). Most of the small RNAs identified were 24-nt heterochromatic siRNAs as expected. In addition, there were a total of 86 miRNAs from 51 distinct miRNA families (Supplementary Data 10). Among these 51 miRNA families, we identified 21 novel miRNAs; 8 of these were 22 nucleotides in length and thus candidate triggers of secondary siRNAs, known as phased siRNAs or 'phasiRNAs'[24] (produced from *PHAS* loci). Four of the 22-nt novel miRNAs, lsa-m1604, lsa-m5672, lsa-m1606 and lsa-m1057, were highly expressed. Combining the predicted miRNAs with the PARE data validated the cleavage of 288 miRNA target genes (Supplementary Data 11). A total of 565 *PHAS* loci were identified by scanning the small RNAs against the lettuce genome, the majority (430) of which were annotated as protein-coding genes (Supplementary Fig. 7 and Supplementary Data 12). *NB-LRR* (NLR) resistance protein-encoding genes were the largest class of *PHAS* loci (246 genes), as in many other eudicots such as *Glycine max* and *Medicago truncatula*[25,26]. The second largest class of *PHAS* loci (116 genes) was kinases, the highest number for kinases detected in all angiosperms analysed so far. The single miRNA trigger of these kinase phasiRNAs was predicted to be the novel miRNA, lsa-m5672. The lettuce-specific miRNA and high number of targets suggests a lineage-specific specialization of this post-transcriptional control of kinase genes.

**Protein clustering and gene family analysis**. Clustering of the predicted proteome of lettuce using OrthoMCL[27] with proteins from nine other published plant genomes clustered 251,776 out of the 325,448 proteins predicted for all 10 species into 28,720 groups, which contained 29,511 (76%) of the lettuce proteins (Supplementary Table 5). Across the panel of 10 genomes, 1,475 ortholog groups had a member from each species (Supplementary Data 13). Calculations on this set of genes using *Vitis vinifera* as the reference revealed that Brassicaceae species have the highest rate of synonymous substitutions[28] (ds = ~4.53), followed by *Mimulus guttatus* (ds = 3.78) and *L. sativa* (ds = 3.12), while the tree species have the lowest ds values (~1.65) (Supplementary Data 14). Most of the groups represented gene families that were shared between species (13,611 groups were present in more than one species and contained more than one gene per species).

Forty of 107 tested groups exhibited significant expansion or contractions in gene copy number across the panel of 10 species (see Methods) (Supplementary Data 15). Many (14 out of 40) had kinase-related activity. Of these 14 kinase-related groups, 8 had the receptor-like kinase (RLK) domain structure[29]. From the total 696 RLKs annotated in lettuce, 655 were grouped into 204 groups; leucine-rich repeat-RLKs[30] are over-represented while the lectin-RLKs[30] are under-represented in lettuce compared to the other nine species (Supplementary Data 13). A total of 372 nucleotide-

binding leucine-rich repeat receptor like (NLR) proteins in 31 groups were annotated in lettuce. Two groups that were present in all 10 genomes were significantly expanded in *L. sativa*, *Citrus sinensis*, *Populus tricocharpa* and *Theobroma cacao*. These include the *Resistance Gene Candidate 1* (*RGC1*) and *RGC21* subfamilies in *L. sativa*[31]. Another 30 NLR groups varied in copy number and presence/absence across species. Two of these groups were significantly expanded in lettuce compared to the other genomes; one encompassed the *RGC16* subfamily, which is the biggest NLR family present in lettuce[31]. In contrast, 10 NLR groups showed expansions in other plant genomes but were single or low copy in lettuce. There were four NLR groups that are specific to *L. sativa* at the stringencies used (see Methods) and together contain 141 NLR proteins, with the two biggest containing mostly *RGC12* and *RGC4* members. Considering all 28,720 clusters, 1,617 clusters containing 7,035 genes (18% of annotated genes) were detected as specific to *L. sativa* at the threshold used (Supplementary Data 13). These *L. sativa*-specific clusters showed a significant enrichment of GO terms related to 'Signaling' (GO:0023052) and 'Response to stimulus' (GO:0050896) with 'Defense response' (GO:0006952) (Fisher exact test and FDR corrected *P*-value 1.5 e − 75) showing the greatest enrichment (Supplementary Data 16). Expansions of NLR genes have been the result of local duplications since 83% of sequences predicted to encode NLRs are within tandem arrays. These data are consistent with higher evolutionary selection pressure for diversification in *L. sativa* of some families of resistance genes compared to other species. In contrast, genes encoding disease response-related proteins such as the Lipoxygenase (LOX) gene family show greater conservation with no expansion at the family level and the structure of the family is conserved across all genomes analysed (Supplementary Fig. 8).

Analysis of Pfam protein domains revealed expansion and diversification of two families related to latex and rubber production. Seventy-eight genes belonging to the Bet_v_1/Major Latex Protein (PF00407/IPR000916) family[32] were found in *L. sativa*; this is significantly more than the average of 34 present in the other nine genomes studied (Supplementary Data 17). Precise function(s) of these proteins is not known but they are present in latex and involved in responses biotic stresses[33]. The Rubber Elongation Factor (REF) (PF05755) family is also expanded in *L. sativa* with 11 members compared to an average 3.6 of paralogs per genome in the non-rubber producing species (Supplementary Data 17). This family is involved in biosynthesis of rubber in *Hevea brasiliensis*[32] and its copy number is related to the overall capability to produce rubber[34]. *H. brasiliensis* has 18 *REF/SRPP's*, the majority of which are in a single cluster[34]. Similarly, in *L. sativa* that also produces high-quality rubber[35] the major portion of this family is located as a tandem array of eight members located in LG9 (Supplementary Fig. 9). Therefore, this family is expanded in both of these taxonomically distinct, rubber-producing species.

**Genome triplication**. Comparisons of genome structure of the published plant genomes and *L. sativa* against *V. vinifera* as a reference showed that chromosome conservation correlated with growth habit and generation time based on the size of the syntenic blocks as expected (Supplementary Data 18). Species with long generation times or vegetatively propagated (*T. cacao*, *Solanum tuberosum*, *P. trichocarpa*, *C. sinensis*) were more syntenic than those with faster life cycles (*A. thaliana*, *Brassica rapa*). The extent of detectable synteny also correlates with the ploidy level. *T. cacao* (2*n*) exhibits more synteny with V. vinifera (2*n*) than *A. thaliana* (8*n*) and *B. rapa* (12*n*). Consistent with this, *L. sativa* (6*n*) showed an intermediate level of synteny to

*V. vinifera*. Visualization of the syntenic hits between these two genomes suggested a whole-genome triplication event in lettuce since divergence from the grape lineage. *V. vinifera* chromosomes 1,3,9,11,15,16 and 17 are replicated almost entirely in three different locations in *L. sativa* (Fig. 3a). Multiple polyploidization events have been described in different Asterid orders (Lamiales[36,37], Solanales[38,39], Ericales[40], Apiales[41] and Asterales[42]). Analysis of mutation rates between pairs of syntelogs are consistent with whole-genome duplication events in the Solanaceae and Lamiaceae that are independent of the event basal to Compositae family, after separation of these Asterid families (Fig. 3b). All these events correlate with phylogenetic relationships between the species based on CEG gene sequences[19] (Fig. 3c). The divergence between *L. sativa* and the six Lamiid species is similar (Fig. 3c); higher similarity was found to *Actinidia chinensis* (kiwi fruit; Ericaceae), as expected as this species is basal to all Euasterids[43].

Analysis of the intragenomic collinearity in *L. sativa* provided further strong evidence that a large proportion of the genome is present in triplicate (Supplementary Fig. 10). Analysis of the distribution of colinear blocks comprising at least five genes detected 10 separate genomic regions that are present in triplicate

in the *L. sativa* genome (Supplementary Fig. 10). Synonymous substitution rates between syntelogs in the triplicated regions differed between the three regions, suggesting two events leading the triplicated state. These may have involved a whole-genome duplication followed by subsequent hybridization to a related diploid species and then genome duplication of the triploid shortly thereafter as has been proposed for *B. rapa*[44] rather than a single event leading to the triplicated state. Analysis of substitution rates also placed these events 40–45 Myr, soon after the estimated origin of the family 50 Myr (ref. 1) (Supplementary Data 19). Divergence rates were used to assign the paralogous regions as products of the first and second duplication events. Although individual paralogous regions varied, overall the three subgenomes detected were similar in total size, repeat content and number of genes; there was no evidence for genomic changes specific to one subgenome (Supplementary Table 6).

Altogether, the detected triplicated regions cover at least 651 Mb of the *L. sativa* genome (26% of the assembly) and contain 11,816 (30%) of all predicted genes (Supplementary Table 6 and Fig. 4). Of the genes within these regions 2,912 (25%) have retained at least one syntenic counterpart. Some types of genes were enriched in triplicated regions; these included genes

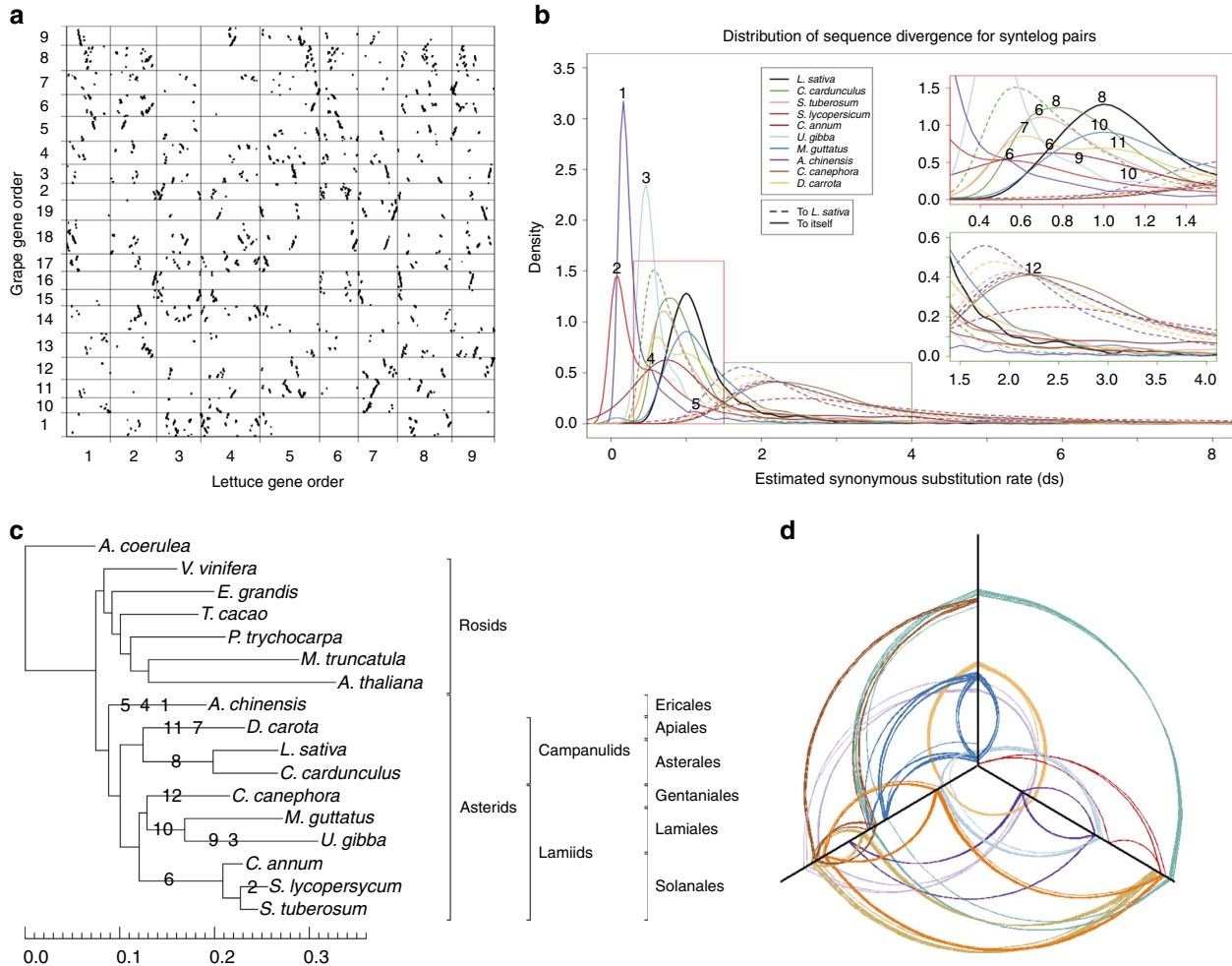

**Figure 3 | Detection of ancient poliploydization events.** (**a**) Syntenic dotplot of *L. sativa* versus *V. vinifera* (*x* axis: *L. sativa* chromosomes; *y* axis: *V. vinifera* chromosomes). (**b**) Density distribution of estimated synonymous substitution rate (ds) of syntelog pairs for intragenomic comparisons and for *L. sativa* against a panel of Asterid species; inserts are enlargements of the main plot. (**c**) RAxML phylogenetic tree of the Asterid clade; scale is estimated nucleotide substitutions per site. Numbers represent inferred positions of the whole-genome duplication/triplication events observed in the syntelogs data (**b**). (**d**) Distributions of triplicated paralogous genes within the lettuce genome. Chromosomal pseudomolecules are arranged progressively along the three axes: *x*: LG1, LG4, LG7; *y*: LG2, LG5, LG8; *z*: LG3, LG6, LG9. WGT, whole-genome triplication.

encoding transcription factors and DNA-binding proteins as well as components of the nucleus, nucleosome, membranes and cell wall. These transcription factors contain a wide set of functional domains (AP2, Homeobox, KNOX, WRKY, TCP) (Supplementary Data 20). Of 54 TCP transcription factors, 27 are present in a triplicated region (Fig. 4); this family includes *Cycloidea*-like genes that have been implicated in the development of the elaborate compound flowers characteristic of the Compositae family[45].

The patterns of retention and diversification differed between families of transcription factors. The nine *Cycloidea*-like genes in lettuce contain the TCP domain, have similarity to genes in other

Compositae species and form a tight clade within the TCP family (Supplementary Fig. 11). Of these, seven are present within the triplicated regions and five have retained syntenic copies. The WRKY family of transcription factors also shows retention of duplicate copies; 32 are present within the triplicated regions and 21 are in pairs or triplets. Clustering of all the WRKY proteins in the genome showed no particular trend in retention across the tree (Supplementary Fig. 12), except as expected syntelog pairs (or triplets) were clustered in most cases. Analysis at the sequence level of these pairs did not provide evidence of diversifying selection; however, the 3′ end was more conserved than the 5′ end with two- to seven-fold differences in pairwise sequence

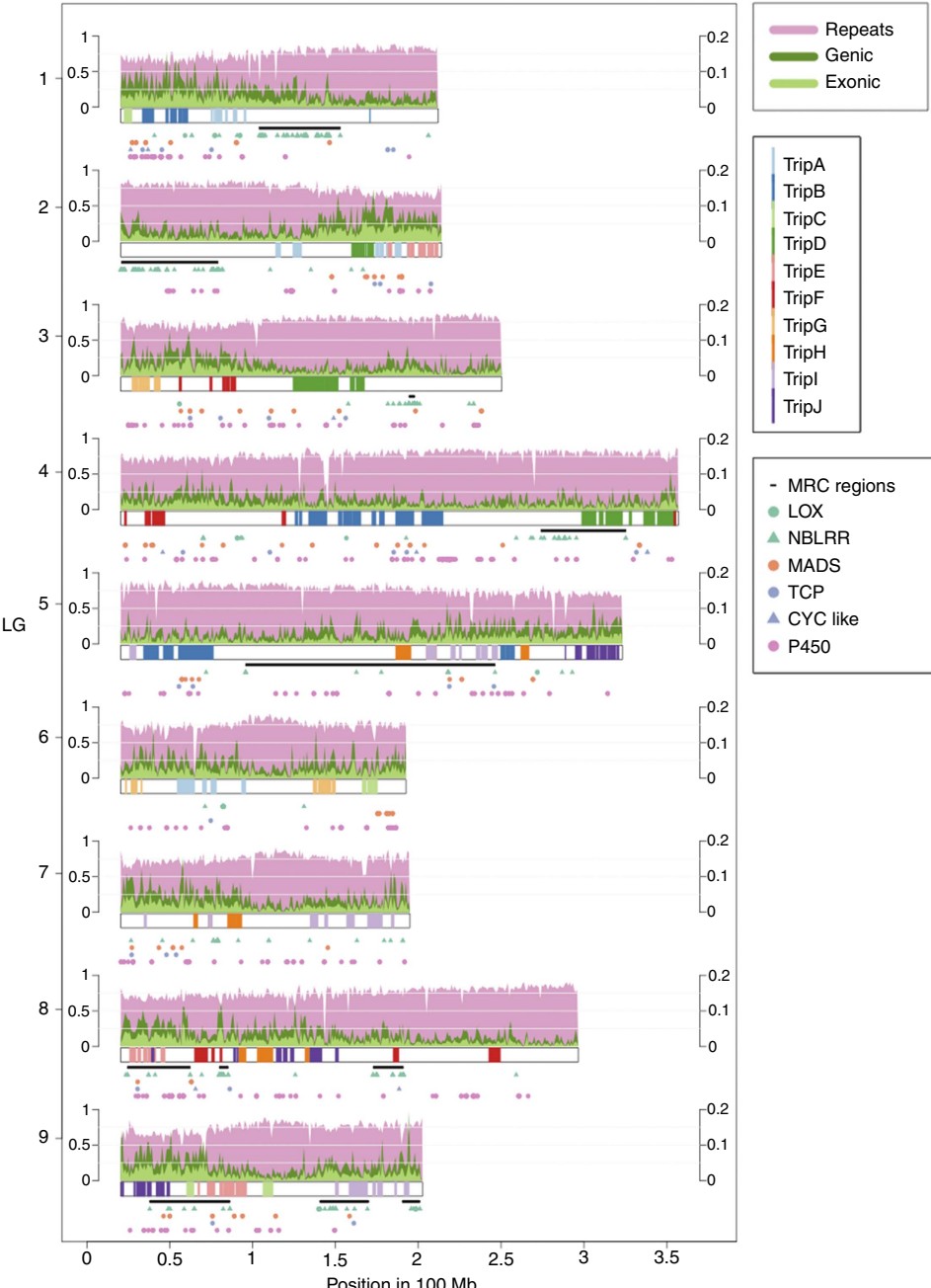

**Figure 4 | Genome-wide distribution of triplicated regions detected in *L. sativa* relative to genomic features and specific gene families.** Each plot represents a single chromosomal pseudomolecule showing the location of the different triplicated regions represented as coloured bars. Repeat, genic and exonic densities are displayed above each chromosomal bar. Due to the large differences in scale, repeat content is scaled on the left axis and genic/exonic content is called on the right axis. Directly below the chromosomal bar are the MRC regions[31]. The lower four tracks show the distribution of different types of genes.

divergence (Supplementary Data 21). This is consistent with selection for conservation of the WRKY domain present in the 3′ end while allowing divergence of the 5′ end potentially leading to gain of functions. Similar analysis of the AP2 family of transcription factors did not detect such differences between the 3′ and 5′ ends (Supplementary Data 21). Expression analysis revealed differences in expression of syntelogs; only 3 out of the 12 WRKY and only 3 out of 23 AP2 syntelog pairs were co-expressed based in a Weighted Gene Correlation-Expression Network Analysis (Supplementary Data 22). Syntelogs were not detected for some genes encoding transcription factor families. The proportion of genes encoding MADS proteins within the triplicated regions (30%) (Fig. 4) is similar to those encoding AP2 (40%) and WRKY (33%) proteins; however, only 6% are detected as syntenic pairs. Since these genes are still present within a syntenic block significant divergence at the sequence level between the paralogs is the likely cause for the lack of detected synteny.

Some genes are under-represented in the detected triplicated regions. Genes encoding proteins related to defence response, signal transduction, protein kinase activity and protein phosphorylation activity are less prevalent in the triplicated regions; only 29 out of 437 NB-LRR candidate genes and 3 out of 21 LOX genes were detected within the triplicated regions. The distribution of these types of genes may reflect the genomic landscape where they are located rather than gene loss itself. MRCs are generally in gene-poor, more dynamic areas with higher repeat content compared to the rest of the genome[31] (Fig. 4). Rapid evolution, higher repeat content and genome rearrangements would cause loss of detectable synteny.

## Discussion

This genome assembly of lettuce is one of the more complete for any plant species reported so far, particularly for genomes larger than 2 Gb with a high repeat content. The genome size of *L. sativa* is typical of many Compositae species[46], despite the family having rapid-cycling weedy species. It provides the first high-quality, genetically validated, reference genome for analysis of the highly successful Compositae family.

The Dovetail technology greatly increased the contiguity of the assembly, assimilated additional scaffolds into chromosomal pseudomolecules, identified chimeric scaffolds that had been missed by genetic analysis, and oriented and ordered scaffolds through complex regions. The Dovetail approach was able to accommodate the high repeat content of the lettuce genome. However, more than half of the remaining scaffolds (6,026) are under 2 kb and predominantly contain repeated sequences; because the HiRise algorithm requires unique alignments of both read pairs such scaffolds are not captured by HiRise. Only a small amount of sequencing was specifically required. The only major requirement is the ability to extract sufficiently large DNA. Therefore, it is readily applicable to numerous species for which there are no or only partial genome assemblies. Consequently, it is likely to result in the rapid increase in the availability of high-quality genome assemblies for numerous non-model species.

The whole-genome triplication event detected in lettuce is basal to the Asterales and distinct from the triplication event recently reported for *Daucus carrota* (carrot) in the Apiales[41], which is also in the Euasterid II clade. Interestingly, these events seem to have occurred simultaneously in common with many lineages at or shortly after the Cretaceous–Paleocene transition (66 Myr), at which time ∼75% of all species became extinct[47]. This is consistent with polyploidy resulting in evolutionary innovations and phenotypic plasticity that conferred selective advantages for

successful colonization of disturbed habitats[48]. As in whole-genome duplications of other plants[49] and animals[50], genes encoding some but not all transcription factors were enriched in triplicated regions of the lettuce genome. Patterns of sequence divergence indicated subfunctionalization at the 5′ end of WRKY genes, and possible neo-functionalization of the AP2 and MADS genes. These changes in regulatory machinery could have led to new phenotypes that would have enabled members of the family to adapt to new environments.

Other genomic features that may have contributed to the success of the family include diversification of *Cycloidea*-like genes that may be involved in formation of the complex capitulum (a composite of many flowers); such an elaborate reproductive organ may be more attractive to pollinators. The Compositae are renowned for their diversity of secondary metabolites[51,52]. Genes involved in the production of latex that gives lettuce its characteristic milky sap were expanded and may be involved in defence against biotic stresses. It is also interesting that there are over 20 novel miRNA, one of which targets kinase transcripts; this may be indicative of another level of regulatory innovation; it will be informative to analyse the miRNA repertoires of other Compositae species.

This high-quality reference genome provides the foundation for syntenic inferences across the Compositae. It will also greatly facilitate map-based cloning of agricultural important genes from lettuce that is required for crop improvement using genome editing, particularly those involved in resistance to abiotic and biotic stresses.

## Methods

**Library preparation and sequencing.** Further details of all methods are presented in Supplementary Note 1. DNA was extracted using a modified CTAB method[53] from seedlings of *L. sativa* cv. Salinas grown in the dark. Seven genomic libraries were constructed with insert sizes from 170 bp to 40 kb and sequenced on the Illumina HiSeq 2000 to generate 298 Gb of raw sequence. Filtering to remove low-quality reads, adapter contamination, ambiguous bases and error correction provided 198.5 Gb of clean data for assembly.

The 99 recombinant lines (RILs) derived from *L. sativa* cv. Salinas × *L. serriola* acc. US96UC23 were a subset of the 213 RILs used to generate Affymetrix GeneChip map[20]. Genomic DNA was isolated from leaves of the RILs using a modified CTAB extraction method[54]. Genomic paired-end libraries were then prepared using standard procedures. Libraries were sequenced to approximately 1x on an Illumina Hiseq 2000.

Two Chicago libraries were prepared by Dovetail genomics as described previously[13]. Libraries were sequenced in two lanes on an Illumina HiSeq 2500 in rapid run mode to generate 313.5 and 357.2 M 100 bp read pairs. This provided a total of 72 × physical coverage.

**Genome assemblies.** SOAPdenovo2 was used for initial assembly of the genomic libraries with a round of gap-filling after the scaffolds had been constructed. This assembly was used as input to the HiRise program along with data from the Chicago libraries for scaffolding using contact frequency information. After scaffolding, another round of gap-filling was done to close the gaps. Two iterations of HiRise scaffolding were done; the first used a single lane of sequencing from one Chicago library and the second used data from sequencing one lane of each two Chicago libraries made from larger DNA fragments. Completeness of the gene-space was evaluated using CEGMA, a set of 357 Ultra Conserved Ortologous Sequences and EST sequences from NCBI for *Lactuca* spp.

**Assembly validation and genetic analysis.** Reads for each of the 99 $F_7$ RILs were mapped to the genomic sequence of *L. sativa* using CLC Genomics Server 6.5 (Qiagen, Redwood City, CA, USA) and haplotypes assigned to each scaffold in each RIL based on the consensus all single-nucleotide polymorphisms detected per scaffold. Scaffolds that exhibited haplotypes with many discontinuities at the same position were considered to be chimeric (Supplementary Fig. 1); these putative chimeras were manually split and the subsections mapped independently.
All scaffolds over 1 kb were clustered into nine chromosomal linkage groups using MadMapper[55]. Scaffolds within each chromosomal linkage group were then assigned to genetic bins based on their segregation using MSTmap[56].
For comparison of the new genetic map with the previous Affymetrix GeneChip map[20] unigenes used in the chip-based map[57] were mapped to the SoapDenovo assembly.

Genetic information from the map based on the SOAPdenovo scaffolds was used for validation of the HiRise superscaffolds by correlating their physical and genetic positions. After validation and clean-up of the HiRise assembly, superscaffolds were mapped into the nine LGs. An initial order was assigned based on the genetic position of the first and last scaffolds within the superscaffold; validated superscaffolds were oriented based on their terminal locations. This order was revised based in the terminal haplotypes to minimize the double recombinants and genetic inconsistencies; the refined order of the superscaffolds was used to construct chromosomal pseudomolecules joining superscaffolds with 10 kb of N's as spacers. Telomeric sequence arrays were found by string searches of known telomere sequences.

**Prediction and analysis of repeat sequences.** Transposable elements were identified in the genome using a combination of homology-based and *de novo* approaches[58]. The genome was mined for repeat elements using ProteinMask and RepeatMasker[59] using Repbase[60] and a set of custom repeat libraries as reference. Additionally, TRF[61] was used to find tandem repeats.

**Prediction of non-coding RNA genes.** Prediction of ncRNA was done by type, and first tRNAscan-SE[62] was used to predict tRNAs. Similarly snoscan[63] and RNAmmer[64] were used to predict snoRNA and rRNA, respectively. Infernal[65] was used for prediction of more ncRNA using the rFam database[66] as input for miRNA, rRNA, rybozimes, snRNA and tRNA.

**Small RNA and target gene analysis.** Total RNA was isolated from leaves of *L. sativa* cvs. Salinas, Cobham Green and Diana, infected or not infected by *Bremia lactucae* (10 samples total). Small RNA libraries were constructed using the TruSeq Small RNA Sample Preparation Kit (Illumina, Hayward, CA, USA). Parallel analysis of RNA end (PARE) libraries were constructed as previously described[23]. The libraries were sequenced on an Illumina HiSeq 2000 at the Delaware Biotechnology Institute (Newark, DE, USA).

The raw reads of sRNA sequencing data were trimmed to remove adaptor sequences and then mapped to the lettuce genome using Bowtie[67]. Reads that matched tRNAs, rRNAs, snRNA and snoRNAs were excluded. Only reads that perfectly matched the lettuce genome were used for further study; miRNA prediction was performed using the previously reported pipeline[25]. PHAS analyses were conducted as described previously[25,68]. Genome-wide miRNA target prediction was performed using the sPARTA package[69] with the built-in target-prediction module miRFerno, followed by PARE-based validation of predicted targets.

**Gene annotation.** Protein-coding genes were predicted using multiple gene annotation pipelines. Two sets of GLEAN predictions and one set of MAKER annotations were combined into a single pool. These data were reduced to non-redundant predictions and further filtered based on an Overlap Weighted Evidence Gene-model (OWEG) score for reliability of the predictions (S.R.-C.-W., unpublished observations). The predicted proteome and genic region were evaluated with CEGMA[19] and against the UCOS[70] to evaluate their completeness. Tandem gene arrays were identified using the CoGe platform as one of the SynMap[71] outputs.

A phylogenetic tree was constructed using sequences of predicted CEG genes using CEGMA[19] on a diverse sample of Eudicot species. Sequences were aligned on Clustal Omega. These alignments were inspected visually, concatenated and used as input into Mega[72] to determine the best substitution model. Concatenated alignment was then used to construct a maximum likelihood tree in RAxML[73].

**Protein clustering and functional annotation.** Functional annotation of all the transcripts was done using InterProScan5[21] and KEGG Automated Annotation Service (KAAS)[74]. Predicted proteins were BLASTed against *A. thaliana* TAIR10 proteins with BLASTp $1 \times e^{-10}$ to find homologous sequences. Analysis of a set of protein families domains from Pfam database v27.0 (ref. 75) was done on the predicted *L. sativa* annotations and a panel of published plant genomes. Number of protein domains detected per species was analysed for abundance of particular gene families in *L. sativa*.

The predicted proteome of *L. sativa* was clustered in OrthoMCL[27] with proteins from published plant genomes. Genes present in clusters that were single copy and present in all genomes were used to calculate divergence with *V. vinifera* as the reference. $X^2$ test was performed in a subset of clusters. Using annotations from *L. sativa* clusters were classified as Receptor Like Kinases (RLKs), Nucleotide binding Leucine rich repeat Receptors (NLRs), lipooxinagenase-like (LOX-like) and subjected to further analysis. Genes present in clusters specific to *L. sativa* were used for GO enrichment analysis using Blast2GO.

**Analysis of synteny.** Synteny was analysed using the CoGe platform (https://genomevolution.org/). *V. vinifera* and *L. sativa* were used as references against 16 publicly available plant genomes including *C. cardunculus*[9]. Besides synteny to the two reference genomes intragenomic synteny was calculated for all Asterid genomes. For comparisons against lettuce and for the intragenomic comparisons, sequence divergence estimates between syntelogs were calculated as part of CoGe.

Triplicated blocks in the *L. sativa* genome were identified by aligning syntenic blocks to the reference and extracting regions that had at least five syntenic genes across two blocks. These blocks were visually inspected and organized into triplicated regions. Divergence between the paralogous regions was calculated using the divergence between the syntelogs within each region. This information was used to group the regions into three subgenomes. Analysis of gene contents of these triplicated regions was done first by GO enrichment analysis using Blast2GO. Additional analysis was done for four families of transcription factors (WRKY, AP2, TCP, MADS). All of the predicted genes in these families were aligned on Clustal Omega and a tree constructed using Clustal Phylogeny. In parallel, mutation rates were calculated for pairs of syntelogs at the whole gene level and also for both termini independently.

Expression data for the genes within the triplicated regions were mined from multiple RNA-seq experiments. These reads were mapped to the genome and read counts were generated for the predicted genes. Downstream analysis was done using R3.1.0 (ref. 76) to normalize the raw counts with the DESeq package and then input to the Weighted Gene Correlation-Expression Network Analysis package for network construction.

**Data availability.** Annotated version of the genome assembly is available from NCBI Genebank under project PRJNA173551 and CoGe[71] (Organism ID 36218, Genome ID 28333). All raw reads for *L. sativa* cv. Salinas have been deposited at the NCBI Sequence Read Archive (SRA) under umbrella project PRJNA173551, accession SRP105354. Sequence for all 99 $F_7$ RILs have been deposited under umbrella project PRJNA243095, accession SRP040749. Small RNA and PARE data have been deposited at the NCBI Gene Expression Omnibus (GEO) under accession GSE84280. Genome browsers and additional information of the assembly and annotation are available at http://lgr.genomecenter.ucdavis.edu. Additional information is available upon request from the corresponding author.

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

## Acknowledgements

We thank Bicheng Yang (BGI) for the project management that generated the initial assembly. We also thank the Dovetail Genomics team, particularly Margot Hartley, Michelle Vierra, Marco Blanchette, Brandon Rice and Nik Putnam, for generating and analysing the *in vitro* proximity ligation data as well as Kariena Dill for comments on a draft of this paper. The work was supported by the Lettuce Genomics Sequencing Consortium that comprised of Agrisemen, BGI, Enza Zaden, Gautier Semences, ISI Sementi, Monsanto, Rijk Zwaan, Syngenta, Takii, Tozers, Vilmorin and The UC Davis Genome Center, plus the NSF Plant Genome Program award # DBI0820451 and the National Research Initiative (NRI) Specialty Crops Research Initiative (SCRI) of the USDA Cooperative State Research, Education and Extension Service (CSREES) awards

# 2010-51181-21631 and 2015-51181-24283 to R.W.M. Analysis of small RNAs and phasiRNAs was supported by NSF IOS award #1257869 to B.C.M.

## Author contributions

The BGI team (Z.W., X.Y., C.S., L.X., S.Z., C.X., X.X.) was responsible for the library construction, genome sequencing, and initial assembly and analyses. The UC Davis team (S.R.-C.-W., D.O.L., A.K., L.F., M.-J.T., H.X., K.C., I.K., R.W.M.) was responsible for material preparation, sequencing of the gene space, cDNAs and RILs, genetic validation, and secondary analyses. Small RNAs were analysed by S.A. and R.X. with input by B.C.M. The paper was written by S.R.-C.-W. and R.W.M. with significant contributions from many of the authors.

## Additional information

**Competing interests:** The authors declare no competing financial interests.

