## [Peer Review File · Nature Communications]

Reviewers' comments:

Reviewer #1 (Remarks to the Author):

The article by Reyes-Chin-Wo et al. describes a new version of the content and structure of the genome of the lettuce *Lactuca sativa*, from the Compositae family. The assembly and scaffolding were performed using a combination of shotgun sequencing, de novo assembly, and scaffolding using a HiC derivative approach developed by the private company Dovetail. This comprehensive approach resulted in a draft genome of improved quality made of 11,474 scaffolds (from 21,686 with shotgun assembly alone). A strong improvement in the assembly results from the addition of linkage analysis of 99 recombinant inbred lines, which allowed the clustering of these scaffolds into nine chromosomal pseudomolecules. The authors have also experimentally characterized and identified small RNAs (though with little functional insight). They discuss gene families, with the genome displaying an enrichment in rubber producing protein encoding genes. Finally, the structure of the genome is refined. The syntenic pattern with respect to the genome of *V. vinifera* as well as an intragenomic colinearity analysis points at a triplication of the genome in the family. The authors finally describe families of transcription factors that have been enriched subsequently to the genome duplications, with putative hints for diversification of some of them (though, again, with no functional insights).

Overall, this is a comprehensive work, and a nice strength of this study consists into the validation of the assembly through a variety of approaches. However this work remains quite similar (in its conclusions and in its aspect) to a study published previously by the same group in 2013 (Truco et al., G3, 2013) that consisted into a high density genetic map of lettuce using the same recombinant inbred lines. That former work notably led to the same conclusions regarding the triplication of the genome in the Compositae family. I am therefore not convinced that the improved assembly led to any new biological result regarding the genome structure, or genomic content, of this species. The title suggests that the identification of a triplication event is new, but it is not that clear. I would be pleased to see a comparison on the maps published in 2013 and the new ones, for instance, to see how much improvements was reached in the present study. As well as a discussion on what's really new here and directly resulting from the better assembly.

Most of the characterization of the nine chromosomes pseudomolecules results from the linkage analysis (which was shown before), and it remains unclear what was exactly the contribution of proximity ligation to the final assembly. The longer superscaffolds generated through this approach are obviously of great help to reorient bins but there is a lot of scaffolds not included in the final assembly, as well as a lot of gaps in the final chromosomes pseudomolecules. The authors state that half of the remaining scaffolds are 2kb or less, but that leaves 6,000 scaffolds of a larger size. What about those? The article is missing a comprehensive diagram or table of what exactly did the Dovetail improved both quantitatively and qualitatively in the final end assembly (and what it would look like without this step). In my eyes, relying on classical genomic metrics such as N50, max size, etc. is not sufficient with this approach and can be misleading, given the lack of understanding of the technique by many scientists. Figure 1 is confusing (and panels on the

bottom seem to overlap somehow). Maybe the author could do a better job at explaining the different steps that leads to their final chromosome macromolecules, with a diagram and info about DNA content at each step? Even though, the gain in N50 is fine but not that impressive (x3.8), and same for the metrics in Sup Table 2. Therefore, the contribution of the Hirise data is of interest, but the title is misleadingly suggesting a greater achievement. In the end, I am a bit surprised and disappointed that the final genome remains a collection of thousands of pieces larger than 100kb, as the writing suggested something bigger was achieved. I understand that the repeats impair for the moment a better assembly, but what are the new biological insights resulting from this one?

Other comments:

In Figure 9 of Truco et al., G3, 2013, the *V. vinifera* genome has 17 chromosomes. In the very similar panel 3A of the present publication, it has 19. Is there a reason for that? How do the two panels compare? Order of chromosomes along the y-axis of Figure 3A is mixed up. Axes could be labeled directly in the figure. In general, many axis legends are too tiny to be read.

The figures overall could be improved for clarity and precision: for instance, on Figure S10, it is unclear which ones are the 10 genomic regions present in triplicate. The author could point at these regions directly on the side of the map with a symbol or stg to help the reader to identify them (even though they are also described in Table S23).

What are the size of the largest scaffolds not included? Would it be possible to visualize the Hirise data for the largest scaffolds not included into the 9 pseudomolecules?

The authors should cite in the discussion or intro the first publications that show that DNA contacts can be exploited to improve genome scaffolding (Burton et al. 2013; Kaplan and Dekker 2013, and MarieNelly et al 2014 for the scaffolding of an incomplete published genome). There is not a big literature on this approach yet so it may guide the reader to a better understanding of the field/technique.

Reviewer #2 (Remarks to the Author):

Lettuce (*Lactuca sativa* L.) is an important vegetable crop species. This manuscript provides a chromosome-scale genome assemble using a variety of approaches including mate-pair data, in vitro proximity ligation, and linkage map. The discovery of genome triplication is important. The paper presents important resource for genome evolution studies of Compositae as well as for lettuce breeding. However, the other parts of the paper are rather routine analyses and have little novelty.

1. Page 1, in the Abstract, the authors claimed "the first comprehensive genome assembly", which I think is not appreciate, as already two Compositae species have been sequenced. The N50 contig size of lettuce genome is 36 Kb, which should be considered as the draft

genome quality. Also over-statement in the Discussion part should be revised.

2. The implication of lettuce genome in Asterid genome evolution can be discussed. Compositae represents a major lineage of Asterid. The divergency of Asterids and Rosids in Eudicot is an important question. I hope the authors can re-do some analyses and present some results on this aspect.

3. To improve the quality of the paper, the authors could consider to perform some biochemistry or molecular biology experiments to investigate the biochemical function of the expanded rubber gene family.

4. In ONLINE METHODS Page 8 line 326 "Expression data for ...was from ### treatments". What was "#"?

Reviewer #3 (Remarks to the Author):

In this manuscript the authors report the creation and analysis of a genome assembly for lettuce. Lettuce is a significant crop species, and also represents a family within plants (the Asteraceae) which is somewhat under represented in terms of reference genome sequences relative to sequence richness. As such, I think this work will be of interested to and beneficial to a wide community of researchers. The fundamental analyses presented here appear sound. I have one moderate concern and several extremely minor points.

1. The authors state on lines 240-243 that they were able to conclude that the WGT in the lineage leading to lettuce resulted from a tetraploidy followed by another duplication based on Ks, but there isn't enough information presented on how they analyzed the data or drew this conclusion to properly review this point in the present manuscript. Without more detail, it's not clear to me how the authors could distinguish their model (duplication of one subgenome within a tetraploid) from the model which is believed to have occurred in the Brassica rapa hexaploidy (a tetraploid species crosses to a diploid relative, creating a triploid which doubles back to a hexaploid, see doi: 10.1534/genetics.111.137349) using Ks data alone.

Minor

2. Lines 90-94: Why was such a low percent identity cut off used when aligning EST sequence data from lettuce back to itself? In other crops I've seen cut offs more in the range of 95% to 98% for percent identity.

3. Lines 122-125: If only 787 super scaffolds have multiple markers not placed into a single genetic bin, how was it possible to orient 859 super-scaffolds using data on within scaffold crossovers?

4. Lines 223-225: Brassica rapa and Arabidopsis thaliana also have more additional whole genome duplications, which tend to shuffle and chop up syntenic regions than do the other species listed in this analysis. For example chocolate is diploid with respect to grape, and poplar is tetraploid with respect to grape, while Arabidopsis is octoploid with respect to grape as a result of two successive whole genome duplications, and in brassica rapa there

are 12 regions equally syntenic and orthologous to any given grape region.

5. For Figure 3B and 3C I suggest using numbers or letters or other symbols. Took me quite a while to convince myself that the specific color or shape assigned to a given ancient polyploid event didn't have any information content.

The authors appear to have done a great job of making their reference genome sequences available through depositions in both NCBI and CyVerse's CoGe.

Response to reviewers NCOMMS-16-19423-T:

Reviewer #1:

The article by Reyes-Chin-Wo et al. describes a new version of the content and structure of the genome of the lettuce *Lactuca sativa*, from the Compositae family. The assembly and scaffolding were performed using a combination of shotgun sequencing, de novo assembly, and scaffolding using a HiC derivative approach developed by the private company Dovetail. This comprehensive approach resulted in a draft genome of improved quality made of 11,474 scaffolds (from 21,686 with shotgun assembly alone). A strong improvement in the assembly results from the addition of linkage analysis of 99 recombinant inbred lines, which allowed the clustering of these scaffolds into nine chromosomal pseudomolecules. The authors have also experimentally characterized and identified small RNAs (though with little functional insight). They discuss gene families, with the genome displaying an enrichment in rubber producing protein encoding genes. Finally, the structure of the genome is refined. The syntenic pattern with respect to the genome of *V. vinifera* as well as an intragenomic colinearity analysis points at a triplication of the genome in the family. The authors finally describe families of transcription factors that have been enriched subsequently to the genome duplications, with putative hints for diversification of some of them (though, again, with no functional insights).

Overall, this is a comprehensive work, and a nice strength of this study consists into the validation of the assembly through a variety of approaches. However this work remains quite similar (in its conclusions and in its aspect) to a study published previously by the same group in 2013 (Truco et al., G3, 2013) that consisted into a high density genetic map of lettuce using the same recombinant inbred lines. That former work notably led to the same conclusions regarding the triplication of the genome in the Compositae family. I am therefore not convinced that the improved assembly led to any new biological result regarding the genome structure, or genomic content, of this species. The title suggests that the identification of a triplication event is new, but it is not that clear. I would be pleased to see a comparison on the maps published in 2013 and the new ones, for instance, to see how much improvements was reached in the present study. As well as a discussion on what's really new here and directly resulting from the better assembly.

We disagree with statements regarding the precedence and similarity to the Truco *et al.* 2013 (<https://www.ncbi.nlm.nih.gov/pmc/articles/PMC3618349/>) paper. The Truco *et al.* paper describes a dense genetic map based on data generated by hybridization to a gene-based Affymetrix chip; 13,000 genes were ordered into chromosomal pseudomolecules. This does not describe the genome sequence in any way. The Truco *et al.* paper mentions a hexaploidization event somewhere in the lettuce lineage, but at that

time this could not be linked to the polyploidization event basal to the Compositae (as cited on line 234) or characterized in detail due to the nature of the partial EST-based data and uncertainty of accurate gene contents and orders. The assembly described in the current paper allowed us to accurately detect the syntenic segments and assign them to the ancient polyploidization events as well as characterize their gene contents and structure in detail. The Truco *et al.* paper has none of this information. The 2013 map was not compared in detail with the new genetic map due to their completely different contents. We did show their collinearity in supplementary figure 3 and state that the new map provides over 97% of the genome assembly in chromosomal pseudomolecules.

Most of the characterization of the nine chromosomes pseudomolecules results from the linkage analysis (which was shown before), and it remains unclear what was exactly the contribution of proximity ligation to the final assembly. The longer superscaffolds generated through this approach are obviously of great help to reorient bins but there is a lot of scaffolds not included in the final assembly, as well as a lot of gaps in the final chromosomes pseudomolecules. The authors state that half of the remaining scaffolds are 2kb or less, but that leaves 6,000 scaffolds of a larger size. What about those? The article is missing a comprehensive diagram or table of what exactly did the Dovetail improved both quantitatively and qualitatively in the final end assembly (and what it would look like without this step). In my eyes, relying on classical genomic metrics such as N50, max size, etc. is not sufficient with this approach and can be misleading, given the lack of understanding of the technique by many scientists. Figure 1 is confusing (and panels on the bottom seem to overlap somehow). Maybe the author could do a better job at explaining the different steps that leads to their final chromosome macromolecules, with a diagram and info about DNA content at each step?

As the reviewer points out the detailed characterization of the triplication event was done using the pseudomolecules since precise syntenic information was required to establish the segment boundaries. Contribution of the Dovetail data was important for this aspect of the paper due to the availability of an accurate chromosome structure (due to the increased contiguity of molecules, reduction in the size of genetic bins, and correct orientation of superscaffolds); we showed and explicitly stated this in Figure 1 and lines 120 to 131. It is impossible to provide more details in the main text within the word limits. The requested additional details were/are provided in the supplementary information (Tables 2 and 3). We understand that Figure 1 was confusing due to its layout and size of the panels, the figure has been revised and panels resized to aid clarity.

Even though, the gain in N50 is fine but not that impressive (x3.8), and same for the metrics in Sup Table 2. Therefore, the contribution of the hirise data is of interest, but the title is misleadingly suggesting a greater achievement. In the end, I am a bit surprised and

disappointed that the final genome remains a collection of thousands of pieces larger than 100kb, as the writing suggested something bigger was achieved. I understand that the repeats impair for the moment a better assembly, but what are the new biological insights resulting from this one?

This is one of the best assemblies for a plant genome of this size. Many recent papers have smaller contig N50s and much smaller scaffold N50s. Compared to most current plant genomes, this assembly of lettuce is one of the best, even though it is one of the largest genomes assembled to date. Of the papers recently published in *Nature Genetics*, only the carrot genome (with an 0.42 Gb assembly that is a sixth the size of our assembly of lettuce) has better statistics. Lettuce has better statistics and a much bigger genome than peanut (1.5 Gb; Bertoli *et al.*, 2016), orchid (1.1 Gb; Cai *et al.*, 2015) and pineapple (0.38 Gb; Ming *et al.*, 2015), all of which are much less scientifically consequential and economically important.

The reviewer states that a 3.8x improvement in N50 is not impressive when Dovetail have reported 90x improvements in N50. However, the degree of improvement is an artifact/consequence of how good the input assembly was; if the input assembly is poor (as it has been for several animal genomes), then the improvement will be large. Also, all publications using Dovetail so far have been on animal genomes that have very different repeat compositions. This paper is the first report of a plant genome assembly that used the Dovetail protocol and the input lettuce assembly was already fairly good after years of refinement prior to use of Dovetail. As the reviewer states, there are few publications describing the application of Dovetail's technology; therefore this is an important publication that could be cited often.

Other comments:

In Figure 9 of Truco *et al.*, G3, 2013, the *V. vinifera* genome has 17 chromosomes. In the very similar panel 3A of the present publication, it has 19. Is there a reason for that? How do the two panels compare? Order of chromosomes along the y-axis of Figure 3A is mixed up. Axes could be labeled directly in the figure. In general, many axis legends are too tiny to be read.

The Truco *et al.* publication dates back to 2013 for which a different, earlier version of the *V. vinifera* genome was available and used. There would be significant overlap between the two panels, especially if lettuce is use as framework of comparison but that was not relevant for the conclusions of this publication. We have revised the axis legends for Figures 1,3,4 and Supplementary Figures 4,5,6.

The figures overall could be improved for clarity and precision: for instance, on Figure S10, it

is unclear which ones are the 10 genomic regions present in triplicate. The author could point at these regions directly on the side of the map with a symbol or stg to help the reader to identify them (even though they are also described in Table S23).

We tried highlighting the triplicated regions but since they are distributed all over the genome the plot became too complex and uninformative. We have adjusted the size of the dots to aid in visualization of the syntenic regions.

What are the size of the largest scaffolds not included? Would it be possible to visualize the Hi-C data for the largest scaffolds not included into the 9 pseudomolecules?

The largest unmapped superscaffold is 2.7 Mb. It is not clear why contact data along such superscaffolds would be informative or useful. They are not part of the pseudomolecules because of the lack of genetic information. These scaffolds contain only 693 out of the 38,919 genes and therefore are less than 2% of the genic regions.

The authors should cite in the discussion or intro the first publications that show that DNA contacts can be exploited to improve genome scaffolding (Burton et al. 2013; Kaplan and Dekker 2013, and MarieNelly et al 2014 for the scaffolding of an incomplete published genome). There is not a big literature on this approach yet so it may guide the reader to a better understanding of the field/technique.

We agree that this was a little cryptic (in the interest of brevity). We have revised the introduction to include more information and references to the *in vitro* ligation approach.

Reviewer #2 (Remarks to the Author):

Lettuce (*Lactuca sativa* L.) is an important vegetable crop species. This manuscript provides a chromosome-scale genome assembly using a variety of approaches including mate-pair data, *in vitro* proximity ligation, and linkage map. The discovery of genome triplication is important. The paper presents important resource for genome evolution studies of Compositae as well as for lettuce breeding. However, the other parts of the paper are rather routine analyses and have little novelty.

This lettuce assembly is the first high quality reference genome for the Compositae family that contains 10% of all flowering plants. Over 98% of genes are present in chromosomal pseudomolecules. Therefore, it can and will serve as the

reference species for this family as this reviewer states. Consequently, this paper will be highly cited. The reviewer correctly says that many of these analyses are routine. We agree but they are required/expected of any genome assembly paper; however, most of this large amount of data is only included as supplementary data and is not part of the main text.

1. Page 1, in the Abstract, the authors claimed “the first comprehensive genome assembly”, which I think is not appreciate, as already two Compositae species have been sequenced. The N50 contig size of lettuce genome is 36 Kb, which should be considered as the draft genome quality. Also over-statement in the Discussion part should be revised.

We were careful with our wording regarding precedence. As we clearly stated in the introduction, draft genomes (both much smaller) for two other Compositae species have been reported. The assembly for *Conya canadensis* (horseweed) is incomplete and highly fragmented; only 72.6% of the conserved CEGMA genes were complete and it has a N50 of 20 Kb, 13,966 scaffolds for a 344.88Mb assembly. No linkage information is included and so this assembly is not informative for synteny analysis. Artichoke (which we assembled) is also highly fragmented and only covers 66% of the genome, with 48% of the estimated genome size anchored into the genetic map. Therefore this paper describes the first high quality assembly that can be used as a reference genome for the Compositae.

2. The implication of lettuce genome in Asterid genome evolution can be discussed. Compositae represents a major lineage of Asterid. The divergency of Asterids and Rosids in Eudicot is an important question. I hope the authors can re -do some analyses and present some results on this aspect.

We agree that Asterid and Rosid divergence needs to be re-evaluated as more Asterid genomes become available. This, however, is beyond the scope of the current paper and was not addressed in this publication due to the low representation of the Euasterid II clade, for which up to now only four, some partial, genomes are available (carrot, lettuce, artichoke and horseweed) compared to the Euasterid I and the Rosid clades. This should be the subject of a future publication when more and better genomes are available.

3. To improve the quality of the paper, the authors could consider to perform some biochemistry or molecular biology experiments to investigate the biochemical function of the expanded rubber gene family.

This is a genome paper that already contains a vast amount of data. Biochemistry or molecular biology experiments to investigate the biochemical function of the expanded

rubber gene family or the unique microRNAs are whole separate studies that would take years to complete.

4. In ONLINE METHODS Page 8 line 326 “Expression data for ...was from ### treatments”. What was “#”?

The text has been revised to correct for this omission. We thank the reviewer for catching this.

Reviewer #3 (Remarks to the Author):

In this manuscript the authors report the creation and analysis of a genome assembly for lettuce. Lettuce is a significant crop species, and also represents a family within plants (the Asteraceae) which is somewhat under represented in terms of reference genome sequences relative to sequence richness. As such, I think this work will be of interested to and beneficial to a wide community of researchers. The fundamental analyses presented here appear sound. I have one moderate concern and several extremely minor points.

1. The authors state on lines 240-243 that they were able to conclude that the WGT in the lineage leading to lettuce resulted from a tetraploidy followed by another duplication based on Ks, but there isn't enough information presented on how they analyzed the data or drew this conclusion to properly review this point in the present manuscript. Without more detail, it's not clear to me how the authors could distinguish their model (duplication of one subgenome within a tetraploid) from the model which is believed to have occurred in the Brassica rapa hexaploidy (a tetraploid species crosses to a diploid relative, creating a triploid which doubles back to a hexaploid, see doi: 10.1534/genetics.111.137349) using Ks data alone.

The reviewer makes a good point and thank him/her for bringing it up. The data indicates that the events occurred within a short time period. On reflection, the scenario suggested by the reviewer is definitely plausible (more likely). We have revised the text appropriately.

Minor

2. Lines 90-94: Why was such a low percent identity cut off used when aligning EST sequence data from lettuce back to itself? In other crops I've seen cut offs more in the range of 95% to 98% for percent identity.

This was used as an inclusive threshold since we report alignments of ESTs from wild

Lactuca species in order to show that we capture the majority of ESTs even from wild *Lactuca* species (Supplementary Table 4). When the identity threshold is increased to >90% and 80% coverage, the proportion of ESTs that mapped to the assembly did not drop much *Lactuca sativa* (94.2% mapped) and *L. serriola* (93.3% mapped). However, at these thresholds the proportion of ESTs from *L. perennis* that mapped dropped to 71.7% as expected.

3. Lines 122-125: If only 787 super scaffolds have multiple markers not placed into a single genetic bin, how was it possible to orient 859 super-scaffolds using data on within scaffold crossovers?

Besides the 787 super scaffolds that span multiple genetic bins, there were 72 that have cross-overs in their terminal sections. These were visually inspected as described in the online methods to determine their orientations. Extra details have been added on the online methods for clarification.

4. Lines 223-225: *Brassica rapa* and *Arabidopsis thaliana* also have more additional whole genome duplications, which tend to shuffle and chop up syntenic regions than do the other species listed in this analysis. For example chocolate is diploid with respect to grape, and poplar is tetraploid with respect to grape, while *Arabidopsis* is octoploid with respect to grape as a result of two successive whole genome duplications, and in *brassica rapa* there are 12 regions equally syntenic and orthologous to any given grape region.

This is an interesting point and is consistent with our data. The text has been revised to include this.

5. For Figure 3B and 3C I suggest using numbers or letters or other symbols. Took me quite a while to convince myself that the specific color or shape assigned to a given ancient polyploid event didn't have any information content.

The figure has been revised accordingly.

The authors appear to have done a great job of making their reference genome sequences available through depositions in both NCBI and CyVerse's CoGe.

REVIEWERS' COMMENTS:

Reviewer #1 (Remarks to the Author):

The revised article by Reyes-Chin-Wo et al. is a slightly modified version of the original manuscript.

The authors argue against some of the comments made during the original review, strongly pointing at the novelty of their results with respect to former work. I have a few comments in this regards.

Firstly, I mentioned Tuco et al 2013 because it seemed to me that one of the major conclusion of the present work regarding the polyploidization event was already made in this former work. However, I agree that the mention to this event was not extensively discussed nor supported, and therefore the present work represents an important advance with this respect.

Secondly, I stated that I was not particularly impressed by the genome assembly improvement. However, I had not realized that the analysis of the lettuce genome was still unpublished in 2016, even without any dovetail assembly step. This is surprising in light of the importance of the lettuce claimed by the authors, and the fact that the genome is available online for a while. Therefore, I agree that overall the results presented in this work present a significant advance compare to what has been published so far.

I agree that the better the original assembly, the more incremental the improvement will be. Plant genomes are particularly difficult to handle, due to their repetitive nature, and here the quality of the assembly is overall remarkable because of all the genetic maps generated in the past. Therefore, I was curious to learn to what extent the improvement brought by Dovetail was essential in bringing the authors to their conclusions. Discussing this may be an interesting way to convince more users to use this technique.

Figure 1. The colors in the expanded view don't match the colors of the top panel, but maybe it is impossible to change do otherwise.

Response to reviewers NCOMMS-16-19423A-Z:

Reviewer #1 (Remarks to the Author):

The revised article by Reyes-Chin-Wo et al. is a slightly modified version of the original manuscript.

The authors argue against some of the comments made during the original review, strongly pointing at the novelty of their results with respect to former work. I have a few comments in this regards.

Firstly, I mentioned Tuco et al. 2013 because it seemed to me that one of the major conclusion of the present work regarding the polyploidization event was already made in this former work. However, I agree that the mention to this event was not extensively discussed nor supported, and therefore the present work represents an important advance with this respect.

Secondly, I stated that I was not particularly impressed by the genome assembly improvement. However, I had not realized that the analysis of the lettuce genome was still unpublished in 2016, even without any dovetail assembly step. This is surprising in light of the importance of the lettuce claimed by the authors, and the fact that the genome is available online for a while. Therefore, I agree that overall the results presented in this work present a significant advance compare to what has been published so far.

I agree that the better the original assembly, the more incremental the improvement will be. Plant genomes are particularly difficult to handle, due to their repetitive nature, and here the quality of the assembly is overall remarkable because of all the genetic maps generated in the past. Therefore, I was curious to learn to what extent the improvement brought by Dovetail was essential in bringing the authors to their conclusions. Discussing this may be an interesting way to convince more users to use this technique.

We agree with emphasizing the significance of the benefits from the Dovetail technology. We have therefore added details in the results section stating this more explicitly (line 152-155).

Figure 1. The colors in the expanded view don't match the colors of the top panel, but maybe it is impossible to change do otherwise.

The reviewer is correct; the colors of the links between the two panels do not match. We also would like them to match but as he points out this not possible. The program used to make the images auto-assigns color when the image is generated and the two images have to be generated independently to avoid pixelation of one or another resulting in degradation of the image.